# The Sts Proteins: Modulators of Host Immunity

**DOI:** 10.3390/ijms24108834

**Published:** 2023-05-16

**Authors:** Anika Zaman, Jarrod B. French, Nick Carpino

**Affiliations:** 1Graduate Program in Molecular and Cellular Pharmacology, Stony Brook University, Stony Brook, NY 11794, USA; anika.zaman@stonybrook.edu; 2Hormel Institute, University of Minnesota, 801 16th Ave NE, Austin, MN 55912, USA; jfrench@umn.edu; 3Department of Microbiology and Immunology, Stony Brook University, Stony Brook, NY 11794, USA

**Keywords:** Sts-1, Sts-2, histidine phosphatase, phosphodiesterase, *Candida albicans*, *Francisella tularensis*

## Abstract

The suppressor of TCR signaling (Sts) proteins, Sts-1 and Sts-2, are a pair of closely related signaling molecules that belong to the histidine phosphatase (HP) family of enzymes by virtue of an evolutionarily conserved C-terminal phosphatase domain. HPs derive their name from a conserved histidine that is important for catalytic activity and the current evidence indicates that the Sts HP domain plays a critical functional role. Sts-1_HP_ has been shown to possess a readily measurable protein tyrosine phosphatase activity that regulates a number of important tyrosine-kinase-mediated signaling pathways. The in vitro catalytic activity of Sts-2_HP_ is significantly lower than that of Sts-1_HP_, and its signaling role is less characterized. The highly conserved unique structure of the Sts proteins, in which additional domains, including one that exhibits a novel phosphodiesterase activity, are juxtaposed together with the phosphatase domain, suggesting that Sts-1 and -2 occupy a specialized intracellular signaling niche. To date, the analysis of Sts function has centered predominately around the role of Sts-1 and -2 in regulating host immunity and other responses associated with cells of hematopoietic origin. This includes their negative regulatory role in T cells, platelets, mast cells and other cell types, as well as their less defined roles in regulating host responses to microbial infection. Regarding the latter, the use of a mouse model lacking Sts expression has been used to demonstrate that Sts contributes non-redundantly to the regulation of host immunity toward a fungal pathogen (*C. albicans*) and a Gram-negative bacterial pathogen (*F. tularensis*). In particular, *Sts*-/- animals demonstrate significant resistance to lethal infections of both pathogens, a phenotype that is correlated with some heightened anti-microbial responses of phagocytes derived from mutant mice. Altogether, the past several years have seen steady progress in our understanding of Sts biology.

## 1. Introduction

The Suppressor of TCR Signaling proteins (Sts), otherwise known as TULA/UBASH3, can be found in an evolutionarily diverse group of organisms, from very primitive metazoans to higher-order Mammalia [1]. Two Sts homologues (Sts-1/TULA2/UBASH3B and Sts-2/TULA/UBASH3A), produced by an apparent gene duplication event, are expressed in higher-order vertebrates (e.g., mammals), while lower-order animals generally express one paralogue. Almost all family members are characterized by a unique structure that combines four distinct protein domains within the same polypeptide: an N-terminal ubiquitin association (UBA) domain, a central Src homology 3 (SH3) domain, a C-terminal histidine phosphatase (HP) domain, and a region spanning the UBA and SH3 domains that contains bipartite 2-H phosphoesterase motifs and exhibits a novel, but poorly characterized, phosphodiesterase (PDE) enzyme activity. Interestingly, the annotated *C. elegans* Sts paralogue appears to lack the UBA, PDE, and SH3 domains, and consist entirely of the isolated C-terminal HP domain [2]. Despite this evolutionary exception, the high degree of conservation exhibited by the full-length protein throughout several hundred million years of metazoan evolution argues strongly for Sts functional importance and the functional importance of each domain. The examination of the murine expression pattern of Sts-1 and Sts-2 has established the relative ubiquity of Sts-1 expression and a generally more restricted expression pattern of Sts-2, with the latter primarily found in cells and tissues of the hematopoietic system [3]. In this review, in addition to highlighting the particularly unique aspects of Sts biology, we will discuss the recent intriguing findings related to the role of Sts in modulating host immunity.

## 2. Results

### 2.1. Sts Catalytic Domains

#### 2.1.1. The Sts Histidine Phosphatase (HP) Domain

The Sts HP domain (Sts_HP_) encompasses approximately half of the Sts polypeptide and places the Sts proteins firmly in a large superfamily of phosphatases, of which the name is derived from a catalytically important histidine within the active site [4]. Despite large differences in their primary amino acid sequences, the overall tertiary structures and substrate specificities, all HPs possess four invariant amino acids (two histidines and two arginine residues) that adopt a signature conformation within the phosphatase active site and are required for enzyme activity [5]. The eponymous histidine serves as the catalytic nucleophile and the primary role of the other three residues appears to be within stabilizing enzyme–substrate interactions [6]. The tertiary structure of Sts_HP_ is an α/β structure resembling other family members [7,8]. Interestingly, E. coli phosphoglycerate mutase (ecPGM) was identified as a close structural homologue of murine Sts-1_HP_, although the active site cleft of the latter appears to be considerably wider and potentially more solvent-exposed than that of ecPGM and other HPs [7]. To some extent, this may account for the ability of Sts-1_HP_ to accommodate large macromolecular substrates (see below).

Histidine phosphatases are categorized into two evolutionary branches, with Branch 1 containing the so-called PGM (phosphoglycerate mutase)-like enzymes and Branch 2 broadly consisting of enzymes known as AcPs (acid phosphatases) [4]. One of the differences between Branch 1 and Branch 2 family members lies in the identity and positioning of a conserved acidic residue that is thought to contribute to the catalytic reaction by acting as a general base. This residue is a glutamate within Branch 1 enzymes and an aspartate in Branch 2 enzymes (Figure 1, top). Another structural difference between the two branches is a conserved arginine residue, of which the side chain projects prominently into the active site cavity (Figure 1, bottom). It is absent in Branch 1 enzymes and highly conserved in Branch 2 enzymes. The function of this arginine is not well characterized, but has been variously ascribed roles in supporting the formation of the enzyme–substrate complex, facilitating the formation of a transition-state structure, or guiding the release of inorganic phosphate from the active site following substrate hydrolysis [9]. Interestingly, the Sts proteins appear to be unique among HPs in possessing elements that distinguish both evolutionary branches of the superfamily, including the catalytic glutamate that characterizes Branch 1 enzymes and the conserved extra active site arginine residue found within Branch 2 enzymes (Figure 1). Whether this structural curiosity is merely an evolutionary oddity or carries wider functional significance requires further investigation.

As reported in the literature, the first substrate used to probe the phosphatase activity of an Sts family member was the conjugated ecdysteroid ecdysone 22-phosphate (E22P), which was also used in an activity assay to purify the Bombyx mori Sts homologue known as EPPase [10]. In addition to hydrolyzing E22P, EPPase demonstrated variable catalytic activity toward 20-hydroxyecdysone 22-phosphate (20E22P), 2-deoxyecdysone 22-phosphate (2dE22P), and 22-deoxy-20-hydroxyecdysone 3-phosphate (22d20E3P). Subsequently, Davies et al. reported a limited E22P activity of human Sts-1 [11]. In a later study, Mikhailik et al. successfully showed that a phospho-tyrosine phosphatase activity of Sts-1 was capable of hydrolyzing pTyr peptides, tyrosine phosphorylated recombinant proteins in vitro, and intracellular tyrosine phosphorylated proteins such as activated pSrc and pZap-70 [7]. The latter example was of great interest because of the appearance of hyper-phosphorylated Zap-70 in T cells lacking Sts expression (see below). Additional tyrosine-phosphorylated proteins shown to be the targets of Sts-1 include the epidermal growth factor receptor (EGFR) and the Zap-70 kinase homologue, Syk [12,13]. The identification of Syk as a bona fide intracellular substrate of Sts-1/TULA-2 was strengthened by results of an in vitro combinatorial Tyr(P) peptide library screen, which identified two distinct classes of Tyr(P) peptide substrates as preferred Sts-1_HP_ substrates. Each class contains a unique set of residues surrounding the pTyr residue, and Tyr(P)-323 and Tyr(P)-352 sites of hSyk were shown to be contained within preferred Sts-1 dephosphorylation motifs [14].

While the phosphatase domain of Sts-1 has potent in vitro protein tyrosine phosphatase activity and is thought to suppress TCR signaling by directly targeting Zap-70, the measurable phosphatase activity of Sts-2 is markedly lower and was not as well established for a number of years [7]. Indeed, in the same combinatorial peptide library screen that identified the pTyr peptide substrate specificity of Sts-1/TULA-2, Sts-2/TULA-1 displayed no hydrolytic activity toward any peptides within the library [14]. However, a wider range of model phosphatase substrates, including 3-O-methylfluorescein phosphate (OMFP) and fluorescein diphosphate (FDP), were used to illustrate a weak catalytic activity associated with Sts-2 [15]. The structural differences between Sts-1 and -2 that account for the substantially different phosphatase activities are not well understood. Despite these differences, however, both Sts-1 and -2 have been shown to play key roles in targeting a T-cell tyrosine kinase (Zap-70) for dephosphorylation and inactivation, as demonstrated partly by an analysis of cells obtained from wild-type, Sts-1^-/-^, Sts-2^-/-^, and Sts-1/2^-/-^ mice [15].

#### 2.1.2. The Sts Phosphodiesterase (PDE) Domain

Through sequence profile analysis, the PDE domain of the Sts proteins was putatively assigned as a member of the 2′, 3′-cyclic nucleotide phosphodiesterase family, a large family of phosphoesterases with two conserved histidines [16]. Although the members of this family have only modest overall sequence similarity, they are unified by the presence of two conserved 2H phosphoesterase motifs, which consist of four amino acids with the sequence Hh(S/T)h, where ‘H’ represents a catalytically important histidine residue and ‘h’ represents any hydrophobic residue [17]. Substrates of 2H phosphoesterase family members include cyclic phosphodiester bonds present on free nucleotides, oligonucleotides, or at the 3’ termini of RNA [18]. Consistent with assignment to the 2H phosphoesterase family, the Sts-1 PDE domain (Sts-1_PDE_) was recently demonstrated to have 2′, 3′-cyclic nucleotide phosphodiesterase activity, and specifically generates the 3′-nucleotide as its product [19]. 

Despite the relatively high variation in the amino acid sequence of the members of this family, the overall structural fold tends to be relatively well conserved, particularly within sub-families [16]. The predicted structure of Sts-1_PDE_ [20] shows the conserved pseudo-twofold symmetrical mixed α/β fold (Figure 2A). Comparison to the most structurally similar sequence characterized 2H phosphodiesterase, YjcG [21], shows the conservation in overall structure, despite low sequence similarity (19% sequence identity, Figure 2B). The putative active site of the protein, as expected, is predicted to be primarily composed of the two 2H phosphoesterase motifs (Figure 2C). A unique feature of the predicted Sts-1_PDE_ structure is the extension of the β-sheet by one strand made up of residues from a sequence-distant part of the protein (Figure 2A, blue). The additional strand is predicted to be composed of residues 317 through 333, while the predicted PDE domain spans residues 87 through 255. While this strand is not necessary for activity [19], it may play a role in organizing the overall structure of Sts-1. 

Through the use of Sts-1 mutants in which critical histidine residues within the 2H phosphesterase motifs were altered to non-functional residues, Yin et al. provided evidence of a functional role for Sts-1_PDE_ in regulating signaling downstream of both the TCR and the fungal receptor Dectin-1 [19]. Further insights into the functional role of Sts_PDE_ await additional biochemical and cell-based analyses.

### 2.2. Functional Studies

Because the Sts proteins were discovered in part by their ability to interact with the E3 ubiquitin ligase, c-Cbl, early functional studies centered on the consequences of the interaction between Sts and Cbl. Kowanetz et al. reported the discovery of UBASH3A/Sts-2 via a yeast two-hybrid screen for Cbl-interacting proteins, and based on studies in HEK293T cells demonstrating that over-expression of Sts-inhibited internalization of activated EGF and PDGF receptors, speculated that the interaction of Sts and Cbl negatively influenced the ability of endocytic sorting proteins to access their substrate receptors [22]. In contrast, Feshchenko et al. isolated TULA/Sts-2 as a Cbl-interacting protein via biochemical means, and proposed that the function of Sts-2 was linked to its ability to induce ubiquitination and degradation of Cbl, thereby down-regulating a protein known to be involved in receptor internalization [23]. Finally, Carpino et al. isolated Sts-1 (designated p70 at the time) in a Jak2 kinase phosphopeptide affinity screen during a search for Jak2-interacting proteins, but no effect of Sts-1 on the regulation of Jak2 functional activity was discerned [24]. A mouse strain lacking Sts-1 expression was also developed, but initial analysis did not yield any clues into Sts function. The development of mice lacking expression of both Sts-1 and -2 (*Sts*-/-) followed shortly thereafter, and this mouse model was used to demonstrate the role for Sts in regulating signaling pathways downstream of the TCR. 

#### 2.2.1. The Role of Sts in T Cells

The generation and analysis of a mouse model lacking the expression of Sts-1 and -2 (*Sts-1/2*-/- or *Sts*-/-) provided initial insights into the role of Sts regulating T-cell activation pathways downstream of the TCR [7]. *Sts*-/- mature thymocytes and splenic CD4^+^/CD8^+^ T cells exhibited a significant hyper-proliferative defect when stimulated in vitro with anti-TCR antibody. Importantly, however, they did not proliferate in the absence of stimulation, and their response to a combination of phorbol ester and ionomycin was identical to wild-type T cells, suggesting a TCR-dependent effect. The stimulation of purified *Sts*-/- splenic T cells with platebound anti-TCR antibody also led to significant increases in cytokine production, including IL-2, IL-4, IL-5, IL-10, and IFNγ, relative to wild-type cells. Finally, antigen-specific *Sts*-/- CD8^+^ T cells derived from influenza A nucleoprotein also displayed an increased expression of IL-2 and IFNγ relative to wild-type cells, following stimulation with NP_366-374_ peptide [7]. These in vitro studies established Sts-1 and -2 as critical regulators of TCR-induced T-cell activation, including T-cell proliferation and cytokine production. As both *Sts-1*-/- and *Sts-2*-/- T cells did not display significantly different responses than wild-type cells following in vitro stimulation, these results suggested Sts-1 and Sts-2 function as overlapping, if not redundant, negative regulators of T-cell activation mediated by TCR stimulation. 

Consistent with the *Sts*-/- T-cell in vitro hypersensitivity phenotype, a subsequent in vivo analysis of naive, unchallenged mice by Newman et al. identified increases in the fraction of splenic T cells exhibiting a mature memory/effector phenotype in *Sts*-/- animals [25]. Interestingly, differences in the *Sts-1*-/- vs. *Sts-2*-/- phenotype were noted, with individual Sts-1 deletion having a more pronounced effect than individual Sts-2 deletion. Through the use of a mouse model of inflammatory bowel disease that is driven in part by activated T cells, it was also demonstrated that the lack of either Sts-1 or Sts-2 alone was sufficient to promote increases in cytokine expression and inflammatory pathology [25]. Further, using an alternative autoimmune model, collagen-induced arthritis, Okabe et al. demonstrated that *Sts-2*-/- mice were significantly more susceptible to autoimmune induction than wild-type or *Sts-1*-/- animals, a finding that correlated with the observation of increased numbers of IL-2-producing splenic CD4+ T cells in *Sts-2*-/- vs. WT or *Sts-1*-/--challenged animals [26]. These studies provided evidence of the possible independent roles for Sts-1 and Sts-2 in regulating in vivo T-cell responses. 

Zap-70, a Syk-family member tyrosine kinase that associates with the CD3ζ subunits of the TCR, plays an indispensable signaling role downstream of the TCR [27]. Analysis of T cells isolated from *Sts*-/- animals revealed Zap-70 as a likely target of Sts phosphatase activity [3]. In particular, the kinase displayed increased stimulation-induced tyrosine phosphorylation and immunoprecipitable in vitro activity in *Sts*-/- T cells, relative to its activation state in wild-type cells. Interestingly, increased tyrosine phosphorylation of the 70 kDa form of Zap-70, as well as additional higher molecular weight forms of Zap-70 corresponding to its ubiquitinated forms, were visible in stimulated *Sts*-/- T cell lysates, although the ubiquitinated Zap-70 displayed no in vitro kinase activity. The signaling elements known to be downstream of Zap-70, such as the adaptor proteins, LAT and Slp-76, were hyper-activated in *Sts*-/- cells, and the levels of phosphorylated MAPK following TCR stimulation were greater and more sustained in *Sts*-/- cells compared to wild-type cells [3]. Thus, the absence of Sts-1 and -2 appears to lead to a generalized and early biochemical defect in early TCR signal transduction, resulting in increases in activated forms of Zap-70. Interestingly, the increased level of tyrosine-phosphorylated, ubiquitinated Zap-70 evident in stimulated *Sts*-/- T cells was later shown to be emblematic of a generalized defect of *Sts*-/- T cells, which display elevated levels of tyrosine-phosphorylated, ubiquitinated proteins following TCR stimulation [28]. 

The factors that regulate Sts expression in T cells remain to be elucidated. Interestingly, a recent report describes the role of PSGL-1 (P-selectin glycoprotein-1), the T-cell intrinsic checkpoint regulator of CD8+ T-cell exhaustion, in maintaining the high levels of Sts-1 expression as one possible mechanism for restraining TCR signaling [29]. Importantly, this is the first study to identify a potential role for Sts-1 in contributing to T-cell exhaustion. 

The early studies described above have led to a model for Sts function in T cells, in which the Sts-mediated dephosphorylation of pTyr substrates (e.g., Zap-70) is spatially and temporally regulated by protein–protein interactions involving the UBA and SH3 domains. Support for this model came from the observation that the Nrdp1 ring-finger-type E3 ubiquitin ligase mediated the polyubiquitination of Zap-70 and promoted the dephosphorylation of Zap-70 by Sts [30]. Additional support derived from functional analysis of a Zap-70 deubiquitinating enzyme, Otud7b, that was shown to facilitate Zap-70 activation by deubiquitinating Zap-70, thereby limiting Zap-70—Sts interactions and the Sts-mediated suppression of Zap-70 phosphorylation [31]. Interestingly, additional negative regulatory functions of Sts in T cells have been proposed, including the role for Sts-2 in suppressing TCR-induced NF-kB signaling via the suppression of IKK activation [32], a role for Sts-2 in promoting downmodulation of TCR-CD3 complexes [33], and the role for Sts-1 in promoting adenosine A2A-receptor-induced downregulation of Notch1 via interactions with the Cbl-b ubiquitin ligase [34]. The mechanistic details of these proposed Sts functions continue to be elucidated.

#### 2.2.2. The Role of Sts in Additional Cell Types

The roles of Sts proteins, primarily Sts-1, in regulating the responses of additional hematopoietic cell types other than T cells have been proposed. For example, Sts-1 is thought to be critical in negatively regulating GPVI-mediated functional responses in platelets through its ability to target the Syk tyrosine kinase [35]. Interestingly, *Sts*-/- animals demonstrate shorter bleeding times and a prothrombotic phenotype [35,36]. A non-redundant role for Sts-1 in negatively regulating Syk activation downstream of FcεRI was also identified following siRNA-mediated knockdown of Sts-1 in a mast cell line [37]. Sts-1 has also been shown to be abundantly expressed in osteoclasts and, based on an analysis of *Sts*-/- mice, it was concluded that Sts-1 negatively regulates osteoclast differentiation and osteoclast activity, likely through its negative regulation of Syk [38]. Overall, *Sts*-/- animals demonstrate a decrease in bone volume that appears to be a result of increased osteoclast numbers and function. Finally, Sts-1 and -2 have been identified as critical regulators of hematopoietic stem and progenitor cell fitness and expansion through their role in negatively regulating FLT3 and c-Kit receptor tyrosine kinases [39]. 

#### 2.2.3. The Role of Sts in Regulating Anti-Microbial Immunity


*Candida albicans*


The role of the Sts proteins in regulating host immunity to microbial pathogens was first examined by testing the susceptibility of *Sts*-/- mice to systemic infection by the dimorphic fungal pathogen, *Candida albicans*. *C. albicans* is a leading cause of hospital-acquired bloodstream infections in the US [40]. In addition, *C. albicans*-related global mortality rates are little changed over the last 25 years [41]. In a well-characterized mouse model of systemic candidiasis that mimics the clinical progression of disseminated candidiasis in humans, progressive sepsis accompanied by renal failure has been identified as the cause of death [42]. Interestingly, *Sts*-/- mice demonstrated profound resistance to *C. albicans* bloodstream infection, with resistance characterized by significantly increased survival, rapid pathogen clearance from the kidney, and an absence of inflammatory lesions [43]. Although the exact cellular mechanisms underlying the enhanced resistance of *Sts*-/- animals to *C. albicans* have not been definitively established, bone-marrow-derived dendritic cells (BMDCs) from *Sts*-/- animals displayed heightened ability to inhibit ex vivo growth of a non-germinating mutant of *C. albicans*. Furthermore, the Syk kinase was hyper-phosphorylated in fungal-stimulated *Sts*-/- BMDCs relative to wild-type cells, and the former cells also displayed a heightened production of reactive oxygen species (ROS) downstream of fungal receptor Dectin-1 [44]. As Sts-2 is downregulated during the differentiation of marrow cells into BMDCs, these findings implicate Sts-1 as a negative regulator of the Dectin-1—Syk tyrosine kinase signaling axis within fungal-stimulated phagocytes. Thus, it is possible that Sts-1 negatively regulates varied innate immune signaling pathways in such a manner that in the context of an infected host, its inactivation promotes enhanced immunological responses and increased resistance to infection. Interestingly, animals lacking Sts-1 or Sts-2 individually displayed significantly increased resistance to low-dose inoculums of *C. albicans*, suggesting that both proteins make important contributions to the regulation of host antimicrobial responses. For high-dose inoculums, an inactivation of both Sts-1 and -2 together are required to observe a significant difference in resistance to infection [43]. Further analysis will be needed to dissect the individual roles of Sts-1 and -2 in the immune response to *C. albicans*.

2.
*Francisella tularensis*


Additional studies have established a role for the Sts proteins in regulating host immunity to a bacterial pathogen, *Francisella tularensis*. *F. tularensis* is a Gram-negative facultative intracellular bacterium and the highly infectious causative agent of tularemia (rabbit fever) [45]. Using a *F. tularensis* live vaccine strain (LVS) that is nonpathogenic in humans, but highly virulent in mice, Parashar et al. demonstrated that *Sts*-/- mice were highly resistant to bacterial infection in a manner that was reminiscent of the response of *Sts*-/- animals to *C. albicans* infection. In particular, mice lacking Sts expression displayed enhanced bacterial clearance in multiple peripheral organs, evidence of reduced inflammation-induced tissue damage, and heightened survival [46]. Interestingly, *Sts*-/- marrow-derived monocytes and neutrophils, but not BMDCs, displayed an enhanced restriction of intracellular bacteria following ex vivo infection. Preliminary evidence suggests that the accelerated bacterial restriction by *Sts*-/- monocytes stems both from the elevated levels of IFNγ produced by infected *Sts*-/- monocyte cultures and an increased responsiveness of the cells to IFNγ that leads to an increased production of IFNγ-induced antimicrobial effectors [47]. The mechanisms by which the Sts proteins regulate IFNγ signaling in the context of a *F. tularensis* LVS infection are not known.

The similarities in the overall response of *Sts*-/- animals to *C. albicans* and *F. tularensis* suggest that the Sts proteins regulate a key host pathway(s) that is important for the innate immunological control of diverse pathogens. Interestingly, the ex vivo infection studies noted above reveal a degree of cell specificity inherent in Sts-regulated responses. For example, for *C. albicans*, *Sts*-/- BMDCs, but not BMDMs or BM-derived monocytes, displayed increases in growth-suppressing activity that appeared independent of any cytokine-induced effector responses [44]. However, in the context of infection by *F. tularensis*, *Sts*-/- BM-derived monocytes, but not BMDMs or BMDCs, demonstrated heightened bactericidal activity that was accompanied by enhanced IFNγ signaling [46]. The underlying origins of this cell specificity are currently unclear, although it suggests the Sts-mediated regulation of intracellular innate immune responses that are uniquely induced within different cell types.

## 3. Discussion

Since the original discovery and cloning of the Sts/TULA/UBASH3 proteins more than a decade ago, our understanding of these unique enzymes and their intracellular functions has grown steadily. A variety of experimental approaches have established them as negative regulators of diverse receptor-mediated signaling pathways, with roles in different cell types, including T cells, platelets, mast cells, diverse phagocytes, and hematopoietic stem cells/progenitor cells. Interestingly, the known targets of Sts phosphatase activity, including the non-receptor tyrosine kinases Zap-70 and Syk and receptor tyrosine kinases such as EGFR and Flt3/c-Kit, are all thought to function at or near the apex of signaling cascades involved in critical cell responses. This suggests that the Sts proteins mainly function within the framework of membrane-proximal regulatory events involved in calibrating the extent to which cells respond to extracellular cues. It will be important to delineate how Sts-1 and Sts-2 are each involved in modulating different cellular responses and the degree to which their individual activities synergize with one another.

Following Sts inactivation, e.g., in the context of siRNA-mediated knockdown or the Sts-/- mouse model, a number of cellular responses are potentiated. For example, T-cells lacking Sts expression demonstrate enhanced signaling responses downstream of the TCR. In this context, it has been hypothesized that the T-cell hypersensitivity resulting from the genetic inactivation of Sts is the most significant underlying component driving increased susceptibility of *Sts*-/- animals to autoimmunity in mouse models of multiple sclerosis (EAE) and colitis (TNBS-induced). Interestingly, a number of GWAS studies have established a link between Sts, particularly Sts-2, and diverse autoimmune phenotypes in the human population [48,49]. However, whether the altered levels of Sts expression associated with Sts allelic variants are a causative factor in autoimmune disease development needs to be more definitely established. Similar to the apparent role of Sts in T cells, current evidence suggests that Sts negatively regulates anti-microbial responses within distinct populations of phagocytes. It is tempting to conclude that the enhanced pathogen clearance and increased resistance of *Sts*-/- animals to *C. albicans* and *F. tularensis* infection is due to the potentiation of anti-microbial responses within the critical phagocyte populations. An important challenge ahead will be to identify the relevant cell populations that underlie the in vivo resistance phenotype, determine the relevant pathways/signaling responses regulated by Sts, and investigate how the modulation of Sts activity impacts host immune responses. 

Additional questions that remain to be addressed are those pertaining to the precise intracellular mechanism(s) of action of the Sts proteins. For example, while it has become appreciated that the Sts histidine phosphatase domain (Sts_HP_) plays a critical functional role, the disparities in catalytic activities between Sts-1_HP_ and Sts-2_HP_ suggest important differences either in the identity of Sts-1_HP_ vs. Sts-2_HP_ intracellular substrates, or in the manner in which the two enzymes engage a common set of substrates. A detailed phosphoproteomic analysis of WT vs. *Sts-1*-/- - vs. *Sts-2*-/- - vs. *Sts-1/*2-/- -stimulated cells (e.g., T cells or monocytes) could shed light onto which individual components of different signaling pathways are regulated predominantly by either Sts-1 or Sts-2. The extent to which the functional properties of the UBA and SH3 domains synergize with their respective HP domain enzyme activities also needs to be resolved. Further investigation into the functional differences between the two HP domains might shed light onto why some cells express both homologues at a high level and other cells express only one family member. 

Finally, another intriguing set of questions involves the functional relevance of the catalytic activity associated with Sts-1_PDE_ and the potential phosphodiesterase activity associated with Sts-2_PDE_, whether Sts_PDE_ targets an intracellular small molecule akin to other 2H phosphoesterase target substrates, and the manner in which Sts_PDE_ engages with intracellular signaling cascades. 

Further insights into these and other unanswered questions will help clarify the signaling role of the unique Sts molecules and determine the feasibility of modulating host immune responses by therapeutically targeting Sts activity.

## Figures and Tables

**Figure 1 ijms-24-08834-f001:**
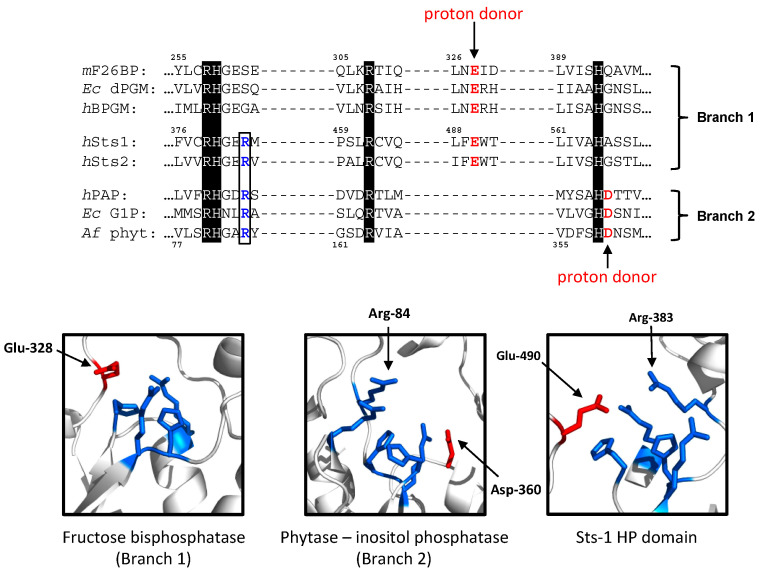
The Sts_HP_ active site contains elements that normally distinguish different evolutionary branches of the HP superfamily. (**Top**): Alignment of Sts-1 and Sts-2 with representative members of Branch 1 and Branch 2 histidine phosphatases. Invariant residues that make up the catalytic quartet are highlighted. For each sequence, the conserved acidic residue thought to participate in catalysis (general base) is indicated in red and the extra active site arginine is in blue. mF26BP (GenBank AAH51014); Ec dPGM: (GenBank HAX1827729.1); hBPGM: (GenBank XP_054214844.1); hPAP: (GenBank XP_011511248.1); EcG1P: (GenBank NP_415522); Af phytase: (GenBank AAB96872). (**Bottom**): Active site of Branch 1 F26P (PDB 1FBT), Branch 2 phytase (PDB 1SKB), and Sts-1_HP_ (PDB 2H0Q) depicting the four invariant residues that comprise the catalytic quartet, the proposed general base, and the extra active site arginine that characterizes Branch 2/Sts enzymes.

**Figure 2 ijms-24-08834-f002:**
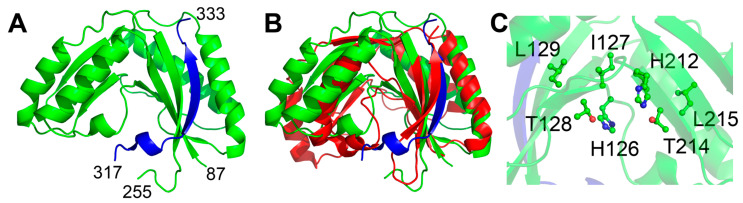
Predicted structure of Sts-1_PDE_. (**A**) The structure of Sts-1_PDE_ (green ribbon), as predicted by AlphaFold, has the conserved pseudosymmetrical α/β fold observed in other members of the family. The β-sheet of one lobe of the protein is predicted to be extended by an additional strand contributed by residues 317–333 (colored blue), which is outside the predicted PDE domain. (**B**) Comparison to B. subtilis, YjcG (2D4G, red), shows the conservation of the overall fold (RMSD 2.75 Å). (**C**) The predicted active site is predominantly made up of residues from the conserved Hh(S/T)h motifs, 126-HITL and 212-HVTL in Sts-1_PDE_.

## Data Availability

No new data were created or analyzed in this study. Data sharing not applicable to this article.

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
