# Peer review of "The Sts Proteins: Modulators of Host Immunity"

_ijms, 2023, doi:10.3390/ijms24108834_

Round 1
Reviewer 1 Report
Dear Authors,
The work is well organized and comprehensively described. The review is characterized by brevity, due to the fact that it covers a rather narrow area of research. In my opinion, it is necessary to expand the Discussion a bit.
The review is devoted to the intracellular functions and mechanism of action of the unique Sts/TULA/UBASH3 enzymes. The review relevant in this field. There are no such reviews in the literature. The authors used a wide set of original material. The conclusions consistent with the evidence and arguments presented. The main task of the review have been resolved. However, the Discussion section may benefit from further expansion. The references cited in the review are appropriate. As for the tables and figures, no additional comments were offered.
Author Response
The Discussion has been expanded by 1) elaborating on some previous points that were not fully discussed; and 2) providing additional sets of comments and discussion that were omitted in the original draft. (Lines 319-366).
Reviewer 2 Report
This paper provides a comprehensive review of Sts proteins and their function in T cells, with a particular emphasis on recent discoveries related to anti-microbial pathogen immunity in Sts-/- mice. The authors are experts in the study of Sts proteins and have provided an accurate and well-covered description of the topic.
However, there are a few areas where the review could be improved. Firstly, there are some mojibakes in Figure 1 that should be corrected. Secondly, while several reviews on Sts/Tula proteins have been cited, it would be helpful for the authors to briefly describe the focus or new contents of the current review compared to others in the introduction section. This would allow readers to easily identify the key newly discovered findings discussed in this review.
Finally, it is worth mentioning a new finding related to Sts-1 that has recently emerged (Hope J. et al. Cell report 2023), which showed that Sts-1 expression is significantly downregulated when PSGL-1 is deficient, which is a discovery that should be added to this review. Overall, this is an informative and well-written review that provides valuable insights into the role of Sts proteins in T cells.
Author Response
- Figure 1 mojibakes: Figure 1 has been corrected to remove mojibakes. A new version of Figure 1 is included in the updated manuscript.
- Brief description of focus/contents: We have included an additional sentence in the introduction to convey the unique aspects of this review relative to previous Reviews. (Lines 52-54).
- PSGL-1 – mediated expression of Sts-1: We have updated the Review to include a recent finding that PSGL-1, a checkpoint regulator of CD8+ T cell exhaustion, promotes expression of Sts-1 (229-233).